# Prevention of Cardiovascular Diseases with Standard-Dose Quadrivalent Influenza Vaccine in People Aged ≥50 Years in Australia During the 2017 A/H3N2 Epidemic

**DOI:** 10.3390/vaccines13040407

**Published:** 2025-04-14

**Authors:** Zubair Akhtar, Aye M. Moa, Timothy C. Tan, Ole Fröbert, Robert Menzies, C. Raina MacIntyre

**Affiliations:** 1Biosecurity Program, The Kirby Institute, Faculty of Medicine and Health, UNSW, Sydney, NSW 2033, Australia; 2Department of Cardiology, Blacktown Hospital, University of Western Sydney, Blacktown, NSW 2148, Australia; 3School of Medical Sciences, Faculty of Medicine and Health, UNSW, Sydney, NSW 2033, Australia; 4Department of Cardiology, Westmead Hospital, Sydney University, Westmead, NSW 2050, Australia; 5Department of Cardiology, Faculty of Health, Örebro University, 701 82 Örebro, Sweden; 6Department of Clinical Medicine, Faculty of Health, Aarhus University, 8000 Aarhus, Denmark; 7Department of Clinical Pharmacology, Aarhus University Hospital, 8000 Aarhus, Denmark; 8Steno Diabetes Center Aarhus, Aarhus University Hospital, 8000 Aarhus, Denmark; 9Sanofi, ANZ, Sydney, NSW 2133, Australia

**Keywords:** influenza vaccination, cardiovascular disease (CVD), hospitalization, vaccine effectiveness, propensity score matching

## Abstract

**Background:** In Australia, 2017 was a severe A/H3N2 season and, therefore, we estimated the effectiveness of standard-dose quadrivalent influenza vaccine in preventing hospitalization for cardiovascular disease (CVD) among New South Wales (NSW) residents aged ≥50 years. **Methods:** We conducted a nested, matched case–control study within the 45 and Up study, linking data from the Australian Immunization Register, NSW Admitted Patient Data Collection and Pharmaceutical Benefits Schedule. Cases were individuals hospitalized for CVD and controls were those who were hospitalized for gastrointestinal diseases. The two groups were balanced using 1:1 propensity score matching based on age group (50–64, 65–74, 75–84, and ≥85 years) and sex. After adjusting for confounders (smoking, body mass index and income), we calculated the adjusted odds ratio (aOR) for vaccination during the season using multivariable logistic regression. E-values were estimated to assess residual confounding. Vaccine effectiveness (VE) was calculated as (1 − aOR) × 100. **Results:** There were 10,445 (4452 cases and 5993 controls) study participants. After matching, 8904 (85.2%) were retained with a mean age of 76.4 ± 10.4 years and 58.3% men. Following adjustment for confounders, the aOR of averting a CVD hospitalization was 0.15 (95% CI: 0.13 to 0.17; *p* < 0.001). The estimated VE against CVD hospitalization was 85% (95% CI: 83 to 87). We found an E-value of 12.82, indicating strong evidence of minimal residual confounding. **Conclusions:** In the severe 2017 influenza A/H3N2 season in Australia, we observed a high VE in preventing cardiovascular hospitalization despite a low VE against influenza infection prevention. Improving vaccine uptake may reduce cardiovascular burden.

## 1. Introduction

Influenza infection affects all ages, with an increased disease burden seen in certain at-risk populations, such as those over the age of 65 and those with chronic comorbidities [1]. In addition, respiratory infections like influenza may have cardiovascular sequelae [2]. Studies have reported an association between influenza or influenza-like-illness and acute myocardial infarction or heart failure in both younger and older adults [3,4,5,6,7]. Mechanisms that have been postulated to explain an increased risk for acute myocardial infarction include the precipitation of plaque rupture [8], endothelial dysfunction [9,10], reactivation of other latent infections leading to plaque rupture [11], fever-associated tachycardia [12], and metabolic derangements related to infection, including elevation of triglycerides and serum glucose levels [13,14]. While the virus is primarily known for causing respiratory distress, it is also linked to high respiratory and cardiovascular hospitalizations, particularly among older adults [15,16,17]. A modeling study in Australia estimated that the average influenza-attributable cardiovascular hospitalization rate in adults >65 years was 57.2 per 100,000 population, whilst cardiovascular mortality was 37.7 per 100,000 population [17].

The 2017 seasonal influenza activity was notably high in the southern hemisphere [18], and in Australia, it was particularly severe, with predominance of the influenza A/H3N2 subtype during that season [18,19,20]. This strain is known to have a more significant impact on older populations [21]. In addition to respiratory morbidity, A/H3N2 has been recognized to have an increased risk of cardiovascular complications, including acute myocardial infarction, myocarditis, and stroke, particularly in vulnerable groups with pre-existing cardiovascular conditions [5,22,23].

Influenza is a vaccine preventable disease and vaccination not only reduces the risk of influenza infection but has also been shown to significantly reduce the incidence of adverse cardiovascular events, particularly among high-risk individuals [24,25,26,27,28]. In a very recent large multinational randomized controlled trial on influenza vaccination in patients with acute myocardial infarction, influenza vaccine reduced all-cause deaths with a hazard ratio (HR) of 0.59 (95% CI, 0.39–0.89) and cardiovascular death HR 0.59 (95% CI, 0.39–0.90) in the 12 months following influenza vaccination [28].

In Australia, influenza vaccine is recommended for all people aged >6 months and provided free of charge under Australia’s National Immunization Program for people aged ≥65 years and for people aged 6 months to 64 years with pre-existing cardiac disease [29]. Quadrivalent influenza vaccines contain two types of influenza A subtypes (A/H1N1, A/H3N2) and both influenza B lineages (B/Victoria, B/Yamagata) [30]. Inactivated quadrivalent influenza vaccines (standard dose) have been available in Australia since 2016 and are licensed for use [31]. In 2017, egg-based vaccines were the exclusive standard dose used in Australia by several manufacturers. The southern hemisphere influenza vaccine used in Australia for the 2017 season included A/Michigan/45/2015 (H1N1)pdm09-like virus, A/Hong Kong/4801/2014 (H3N2)-like virus, B/Brisbane/60/2008-like virus (of the B/Victoria/2/87 lineage), and B/Phuket/3073/2013-like virus (of the B/Yamagata/16/88 lineage) [32]. But, in 2017, there was a mismatch of the A/H3N2 and circulating strains resulting in an overall vaccine effectiveness (VE) against influenza of 33% (95% confidence interval (CI): 17 to 46), 50% (95% CI: 8 to 74) for A/(H1N1)pdm09, 10% (95% CI: −16 to 31) for A/H3 and 57% (95% CI: 41 to 69) for influenza B [18].

The role of the influenza vaccine in preventing cardiovascular disease (CVD) complications during a severe influenza season when its effectiveness against infection is low has not been adequately studied. The 2017 influenza season was particularly severe, making it an ideal opportunity to assess the vaccine’s effectiveness against serious outcomes, particularly CVD hospitalizations. Given that quadrivalent influenza vaccines have been shown to have superior efficacy compared to trivalent vaccines [33], both trivalent and quadrivalent influenza vaccines are recommended for use in Australia, and it is anticipated that by 2026, only the trivalent vaccine will be exclusively used in the country [31]. The World Health Organization (WHO) has recommended using the trivalent vaccine for the 2024–2025 Northern Hemisphere flu season [34]. This recommendation is expected to be rolled out for the 2025 Southern Hemisphere influenza season and beyond. In 2018, Australia’s National Immunization Program also introduced enhanced vaccines, including adjuvanted and high-dose formulations, for adults aged 65 and older [35]. All these changes regarding type of influenza vaccine use marked 2017 as the final year of exclusive use of the standard-dose quadrivalent influenza vaccine in Australia [31], providing a unique opportunity to evaluate its effectiveness before policy changes were implemented.

The 2017 season was particularly valuable for research as it allowed for the assessment of the quadrivalent standard-dose vaccine’s protective effect against CVD hospitalizations during a high-intensity influenza season. Despite the vaccine’s lower effectiveness in preventing influenza infection in 2017 [18], examining its impact on reducing CVD-related hospitalizations can provide crucial insights into the vaccine’s overall public health benefits. Thus, this study aimed to estimate the effectiveness of the standard-dose quadrivalent influenza vaccine in preventing cardiovascular hospitalizations in Australian adults during the 2017 season.

## 2. Methods

### 2.1. Study Setting and Design

New South Wales (NSW) is the most populous state in Australia, with a population of about 8.4 million (31% of the national total) [36]. It includes Australia’s largest city, Sydney, and includes populations in all areas from major cities to very remote areas. The 45 and Up Study is a cohort study of approximately 260,000 residents of NSW aged ≥45 years recruited from a random selection of ~11% residents in that age range [37]. The study commenced in 2006 and was funded by the NSW Department of Health, the Cancer Council NSW, and the National Heart Foundation of Australia (NSW Division). It was designed as a resource for researchers for future research projects. Our study was a nested case–control study design within the 45 and Up Study that included participants who were aged ≥50 years in 2017 and residing in NSW.

### 2.2. Data Collection and Linkage

Participants were members of the 45 and Up study who were aged ≥50 years as of 1 January 2017. NSW Admitted Patient Data Collection data, 01 May–30 September 2017 from NSW Health, Pharmaceutical Benefits Schedule data and the Australian Immunization Register data of 2017 were accessed from the Sax Institute via a secure unified research environment (SURE). In the NSW Admitted Patient Data Collection data, cases were those hospitalized with a principal diagnosis of CVD (I00-I99), and controls were those hospitalized for a principal diagnosis of gastrointestinal disease (but not CVD). The rationale for choosing the control was based on the clinical overlap in presentation between CVD and certain gastrointestinal conditions, particularly in older adults. Both CVD (e.g., acute myocardial infarction, angina) and acute GI diseases (e.g., gastritis, peptic ulcer, pancreatitis) can present with upper abdominal or lower chest pain, and in some cases, distinguishing between the two requires diagnostic evaluation in a hospital setting. As such, patients of both cases and controls are likely to represent individuals with similarly severe symptoms requiring hospital admission, and they may have undergone similar pathways of care, including emergency department presentation and diagnostic workup in contrast to alternatives such as treatment as outpatients or patients admitted for elective surgery. These eligible cases and controls were identified and linked with the 45 and Up study.

### 2.3. Data Transfer and Security

The Centre for Health Record Linkage (CheReL) Unit and 45 and Up Coordinating Centre conducted the data linkage following standard models [38,39]. First, they uploaded the de-identified NSW Health dataset with Project Person Number or unique IDs that were linked to the eligible members of the 45 and Up participants within SURE (a virtual workspace) to conduct the selection of potential participants [40]. The study team checked for the eligibility of potential participants in the study. Eligible 45 and Up study participants were then selected and linked with datasets including the Pharmaceutical Benefits Schedule data. For influenza vaccination records in 2017, the Australian Institute of Health and Welfare provided the Australian Immunization Register data to the Sax Institute to link eligible 45 and Up participants with their vaccination records in the Australian Immunization Register. The Sax Institute mapped and prepared the linked dataset and transferred it to the project’s SURE workspace.

### 2.4. Exposure and Outcome Measures

The exposure of interest was the receipt of influenza vaccine (standard dose) between 01 April 2017 (one month prior to any hospitalization from 01 May 2017) and 30 September 2017. The outcome (cases) was cardiovascular hospitalization from 01 May–30 September 2017. Influenza VE against cardiovascular hospitalization during influenza season from May to September 2017 was estimated in the selected cases and controls.

### 2.5. Data Analysis

The study participant recruitment is described in Figure 1. From the de-identified CHeReL dataset of 266,509 participants, eligible participants were identified based on their hospitalization history in 2017 as cases and controls and selected using the study selection criteria. The total number of eligible participants was 10,465: 4472 cases with CVD admissions and 5993 controls with gastrointestinal admissions. We extracted 2017 influenza vaccination records from the Australian Immunization Register and linked them to our eligible cases and controls for analysis, comprising 10,445 participants (4452 cases and 5993 controls).

Considering the large sample size with sufficient statistical power [41], we opted for a paired matched ratio of 1:1 for this case–control study. We employed a 1:1 propensity score matching strategy to balance the differences between the cases and controls based only on their age groups (50–64 years, 65–74 years, 75–84 years and ≥85 years) and sex (male and female).

This approach was undertaken to minimize potential biases in participant selection and enhance the comparability of the cases and controls. We applied the nearest neighbor matching approach and propensity scores were computed for probability to identify the most similar individuals as cases and controls. A post-matching sensitivity analysis was conducted to ascertain the standardized percent bias reduction. Following propensity score matching, 8904 (85.2%) participants were considered (4452 cases and 4452 controls) for analysis.

Descriptive statistics were conducted for all variables of interest in the final selected matched cases and controls. Missing values that constituted less than 3% were not reported. Univariate analysis by logistic regression was conducted to determine odds ratios (ORs). Following matching for age group and sex only, to account for potential residual confounders, we adjusted for known confounders associated with CVD and vaccination, such as smoking, body mass index (BMI) and income, using multivariable logistic regression to determine the adjusted odds ratios (aORs). The self-reported variables for pre-existing CVD and past cardiovascular admission were excluded as confounders in the multivariable logistic regression due to concerns about reporting bias and recall bias. Subsequently, using the formula “(1 − *aOR*) × 100”, the VE was estimated against cardiovascular hospitalization during the 2017 influenza season. A sensitivity analysis was performed to assess the magnitude of unmeasured confounding of other potential variables by estimating E-values based on the aORs obtained from our multivariable logistic regression model [42,43]. This method determines the minimum association, on the odds ratio scale, an unmeasured confounder must have with both the treatment and outcome to nullify an observed statistically significant effect, thereby addressing any potential residual confounding.

All data were managed by SAS, Version 9.4. (SAS Institute Inc., Cary, NC, USA) and Stata v.18 (StataCorp LP, College Station, TX, USA) was used for descriptive and exploratory analyses.

## 3. Results

The socio-demographic and clinical characteristics of the matched cases and controls are shown in Table 1. There were 8904 cases and controls identified with a mean age of 76.4 ± 10.4 years and 58.3% of them were men. Most of the participants (33.6%) were from the age group of 75–84 years and most (69.9%) of them were born outside Australia. Although 68.6% reported a history of cardiac disease, including hypertension and cerebrovascular diseases, only 37.0% reported having a history of CVD-related admission in the past. Most of the participants (47.5%) reported that they never smoked but among the cases, 40.5% reported they were former smokers and 6.9% reported they were current smokers. The controls had a higher proportion of influenza vaccination (60.7%) compared to the cases (33.0%).

Figure 2 reports the unadjusted and adjusted risk factors associated with cardiovascular hospitalization among the eligible participants. In univariate analysis, being obese, being a current smoker and having an annual income of less than $70,000 Australian dollars (AUD) had higher odds of cardiovascular admission. Participants with influenza vaccination had lower odds (OR 0.32, 95% CI: 0.29 to 0.35, *p* < 0.001) of cardiovascular admissions in 2017. In multivariate analysis, after adjusting for confounders, influenza vaccination was found to be significantly protective with an aOR of 0.15 (95% CI: 0.13 to 0.17, *p* < 0.001). The estimated VE of influenza vaccine against CVD admission was 85% (95% CI: 83 to 87) in 2017 among NSW residents aged ≥50, who were members of the 45 and Up study.

The post-matching sensitivity results demonstrated excellent balance for the matching variables, age group and sex. The standardized bias was 0.0%, *p*-values were 1.000, and variance ratios were 1.00, all falling within accepted thresholds. Overall balance statistics further supported the quality of matching, with pseudo-R² = 0.000, mean and median standardized bias = 0.0%, B = 0.0%, and R = 1.00. The E-value, based on the adjusted odds ratio (aOR) of 0.15 for influenza vaccination and cardiovascular admission, was 12.82, indicating there was no residual confounding by any unmeasured variables.

## 4. Discussion

We found that influenza vaccine was highly and significantly protective against cardiovascular-related admissions during the severe 2017 A/H3N2 influenza season, despite vaccine mismatch and low VE against influenza that year. The 2017 influenza season was a severe season in Australia with a dominant influenza A/H3N2 strain and B lineages in circulation, resulting in high influenza-related morbidity and mortality [17,20]. Consequently, the influenza-attributable cardiovascular hospitalization rate in adults >65 years was 57.2 per 100,000 population, whilst cardiovascular mortality was 37.7 per 100,000 population [17]. The high mobility and mortality rates may attributed to the fact that the A/H3N2 strain has been known to have a more profound effect on older populations [21], increasing their risk of cardiovascular-related complications, especially those with pre-existing cardiovascular complications [5,22,23]. In 2017, approximately 2.2 million doses were recorded in the Australian Immunization Register nationally, including 615,803 doses of quadrivalent influenza vaccines among adults in NSW. The uptake of influenza vaccine in adults aged 50–64 years in 2017 was 10.5% and in people ≥65 years was 31.5% [44]. Our results align with previous studies in other settings that showed 87% VE (95% CI: 35 to 97) in reducing emergency hospitalization for acute coronary syndrome [45] and 80.3% VE (95% CI: 36.3 to 93.9) in reducing hospital admission for exacerbation of chronic heart failure or chronic obstructive pulmonary diseases [46]. A large RCT showed that influenza vaccine significantly reduced all-cause death, myocardial infarction, or stent thrombosis at 12 months [28]. However, this study uniquely reports the VE of influenza vaccine in reducing cardiovascular hospitalizations in 2017 during a A/H3N2 epidemic with low VE against influenza.

Influenza vaccines may have variable efficacy in preventing influenza, depending on the viral strain composition match with circulating strains [47,48]. Also, host factors like “immune imprinting”, defined as the effect of a first infection in an individual imprinting immune response to subsequent infections [49], repeated vaccinations over previous seasons [50], and considering an optimal timing of vaccination into the season play an important effect on the VE of influenza vaccines [51]. These factors affecting the VE have mostly been studied for influenza infection; however, vaccination may still prevent complications of influenza, hospitalizations and deaths in those who do become infected. The high VE in preventing cardiovascular hospitalization in our study may be attributed to the ability of vaccination to prevent complications of influenza, even if influenza itself is not prevented [18]. Most vaccines share this characteristic, including COVID-19 vaccines, where immune evasion has resulted in lower VE against infection, but high VE in prevention of hospitalization and death [52,53,54].

Extensive research has documented that influenza vaccination, in addition to its therapeutic benefit of preventing infection, can effectively prevent adverse cardiovascular events [24,25,27,28]. Numerous studies have evaluated the impact of influenza vaccination on cardiovascular outcomes within the general population and consistently reported a reduced risk of hospitalization due to acute coronary syndromes in vaccinated persons compared to non-vaccinated persons [45,55,56]. A recent review article highlighted the pleiotropic effects of influenza vaccination [57], summarizing numerous studies demonstrating influenza vaccine’s immunomodulatory impact on the innate immune system. The postulated mechanisms include the upregulation and downregulation of several cytokines, conceivably accompanied by cross-reactivity, bystander activation of the adaptive immune system characterized by antigen-independent activation of B and T lymphocytes and influenza vaccine-induced epigenomic remodeling of the innate immune system [57]. Thus, there is ample evidence of various other mechanisms that can lead to a protective effect of influenza vaccination in patients with CVD. However, the primary protective effect of influenza vaccine is to prevent an influenza infection that may lead to a cardiovascular sequelae [2].

The strength of our study is that it is a large observational study examining the effectiveness of influenza vaccine against cardiovascular hospitalizations using linked population data in Australia. Our matched participants comprised many individuals with similar characteristics of age and sex among cases and controls, thus allowing the nested case–control study to control for their confounding effects. Despite its strengths, our study also has some limitations. Although our finding reports VE against CVD hospitalization, we could not examine the history of laboratory-confirmed influenza infection or calculate the VE against influenza in the study population. However, other studies of the 2017 season have done so [17]. Also, we could not determine the temporal association of CVD events and the severity of the illness episode with the effect of vaccination. Another limitation was that 2017 was only the second year of the Australian Immunization Register, and hence vaccination records may have been incomplete. But this is unlikely to have caused substantial bias because those incomplete records were excluded from the analysis. Also, in the 45 and Up study, some variables related to medical history, including asthma and CVD risk factors, were self-reported, potentially leading to reporting bias or recall bias. Consequently, those variables were excluded as confounder variables in the multivariable model. The E-value generated in the sensitivity analysis was large enough (12.82) to reasonably conclude statistically that there was no residual confounding by those unmeasured variables (Appendix A) on the effect of vaccination on cardiovascular hospitalization. However, from a clinical or epidemiological perspective they might have influenced our estimates. Lastly, caution should be exercised in generalizing our findings, as influenza VE may vary by season, circulating strains, vaccine coverage and study populations across different settings.

## 5. Conclusions

Despite low VE against the A/H3N2 strain and B lineages in 2017, our findings demonstrated significant protection of influenza vaccination against cardiovascular hospitalization, consistent with the ability of vaccination to reduce complications of influenza even when infection is not prevented. It is imperative to acknowledge the additional cardioprotective benefits of influenza vaccination and ensure it is translated into policy and practice as a routine preventive measure for cardiovascular diseases, particularly for acute coronary syndrome [58]. With high influenza vaccine uptake in target populations, the additional burden of CVD may be substantially mitigated.

## Figures and Tables

**Figure 1 vaccines-13-00407-f001:**
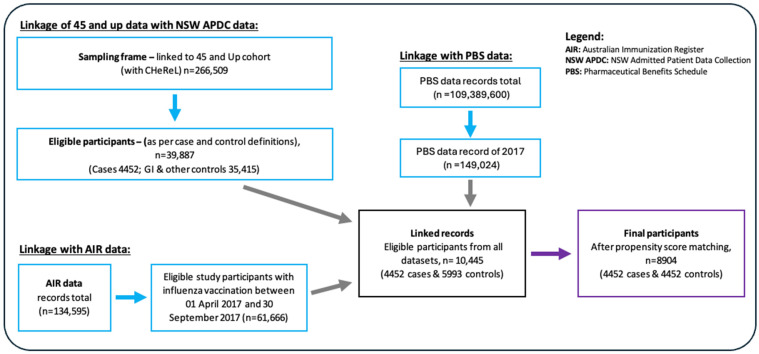
The flow of participant selection for the study.

**Figure 2 vaccines-13-00407-f002:**
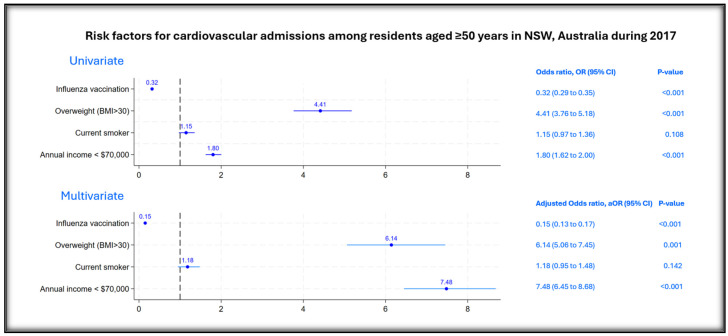
Risk factors for CVD hospitalization among the matched study participants.

**Table 1 vaccines-13-00407-t001:** Sociodemographic and clinical characteristics of matched study participants.

Characteristics	Cases (n = 4452)	Controls (n = 4452)	Total (n = 8904)	*p* Value
Median years (IQR, ±SD)	78 (70–85, ±10.2)	76 (65–89, ±10.6)	76 (68–86, ±10.4)	<0.001
Age groups, years				
50–64	582 (13.1)	582 (13.1)	1164 (13.1)	1.000
65–74	1167 (26.2)	1167 (26.2)	2334 (26.2)	
75–84	1494 (33.6)	1494 (33.6)	2988 (33.6)	
≥85	1209 (27.2)	1209 (27.2)	2418 (27.2)	
Sex				
Male	2596 (58.3)	2596 (58.3)	5192 (58.3)	1.000
Female	1856 (41.7)	1856 (41.7)	3712 (41.7)	
Body mass index, BMI				
Below 25	1197 (26.9)	887 (19.9)	2084 (23.4)	<0.001
25–30 (overweight)	1766 (39.7)	3315 (74.5)	5081 (57.1)	
Above 30 (obese)	1489 (33.5)	250 (5.6)	1739 (19.5)	
Country of birth *				
Australia	1076 (24.2)	901 (20.2)	1977 (22.2)	<0.001
Others	3326 (74.7)	2897 (65.1)	6223 (69.9)	
Aboriginal or Torres Strait Islander *				
Yes	35 (0.8)	0 (0)	35 (0.4)	<0.001
No	4305 (96.7)	4452 (100.0)	8757 (98.4)	
Education *				
Certificate or lower	3541 (79.5)	4202 (94.4)	7743 (87.0)	<0.001
University or higher	824 (18.5)	250 (5.6)	1074 (12.1)	
Income in Australian dollars *				
<70,000	2761 (62.0)	2616 (58.8)	5377 (60.4)	<0.001
>70,000	742 (16.7)	1267 (28.5)	2009 (22.6)	
Prefer not to answer	717 (16.1)	569 (12.8)	1286 (14.4)	
Smoking status *				
Never smoker	2338 (52.5)	1894 (42.5)	4232 (47.5)	<0.001
Former smoker	1801 (40.5)	2226 (50.0)	4027 (45.2)	
Current smoker	308 (6.9)	332 (7.5)	640 (7.2)	
History of past cardiovascular-related admission				
No	2044 (45.9)	3565 (80.1)	5609 (63.0)	<0.001
Yes	2408 (54.1)	887 (19.9)	3295 (37.0)	
Health insurance				
No private insurance	1669 (37.5)	2466 (55.4)	4135 (46.4)	<0.001
Private insurance	2783 (62.5)	1986 (44.6)	4769 (53.6)	
Chronic diseases ᶷ				
Cardiovascular diseases ˠ	2911 (65.4)	3195 (71.8)	6106 (68.6)	<0.001
Hypertension	1638 (36.8)	1589 (35.7)	3227 (36.2)	<0.001
Diabetes	669 (15.0)	569 (12.8)	1238 (13.9)	0.002
Asthma or hay fever	559 (12.6)	0 (0)	559 (6.3)	<0.001
Anxiety and depression	679 (15.3)	482 (10.8)	1161 (13.0)	<0.001
Skin cancer	296 (6.7)	1383 (31.1)	1679 (18.9)	<0.001
Other cancers	601 (13.5)	654 (14.7)	1255 (14.1)	<0.001
Medications ᶷ				
ASA	1386 (31.1)	654 (14.7)	2040 (22.9)	<0.001
Beta blockers	333 (7.5)	0 (0)	333 (3.7)	<0.001
ACEI or ARB	1338 (30.1)	1209 (27.2)	2547 (28.6)	0.646
Statins	1537 (34.5)	1254 (28.2)	2791 (31.3)	<0.001
Influenza vaccination				
Not vaccinated	2985 (67.0)	1749 (39.3)	4734 (53.2)	<0.001
Vaccinated	1467 (33.0)	2703 (60.7)	4170 (46.8)	

* Contains missing values; ᶷ Multiple responses, ˠ any form of cardiac diseases including hypertension and cerebrovascular diseases. Abbreviations: ASA, acetyl salicylic acid; ACEI, angiotensin-converting enzyme inhibitor; ARB, angiotensin receptor blockers.

## Data Availability

Details of the data access policy and procedures are available at www.saxinstitute.org.au. The de-identified, linked health data for this study was accessed within the secure unified research environment (SURE)—a virtual workspace—and only the aggregated findings were disseminated. The SURE workspace was used remotely via the UNSW or UNSW-supported secure pathway. SURE is operated by the Sax Institute and is supported by the Population Health Research Network, which is an initiative of the Australian Government National Collaborative Research Infrastructure Strategy (NCRIS) and the NSW Government.

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
