# Peer review of "Prevention of Cardiovascular Diseases with Standard-Dose Quadrivalent Influenza Vaccine in People Aged ≥50 Years in Australia During the 2017 A/H3N2 Epidemic"

_vaccines, 2025, doi:10.3390/vaccines13040407_

Round 1

Reviewer 1 Report

Comments and Suggestions for Authors

This is an important article showing benefit of vaccines to the general population. 

I have two questions to the authors:

  1. I suggest a paragraph explaining why patients with GI diseases were chosen as control. I understand that a key problem of the case-control studies is how to choose 'cases'.
  2. How authors explain the tremendous difference in some prevalent diseases listed in Table 2. Skin cancer/melanoma affected 100% of patients with GI diseases! I think there is a mistake here. The difference was also strange and related to asthma.

Finally, the exposure variable (influenza vaccination) was done from April to September 2017. Outcomes were recorded throughout 2017 (I think from January 1st to December 31). Are these data correct? If these data are correct, please justify why outcomes were collected before the exposition in the discussion section. 

Reviewer 2 Report

Comments and Suggestions for Authors

I was invited to revise the paper entitled "Prevention of cardiovascular diseases (CVD) with standard-dose influenza vaccine in people aged ≥50 years in Australia during the 2017 A/H3N2 epidemic". It was a retrospective cohort study aimed to evaluate flu vaccine effectiveness in preventing CVDs hospitalizations. 

Despite the interesting aim,methodology was not clear.

Authors stated it was a cohort study with patients selected by the presence/absence of the exposure (flu vaccinations). In the other hand, Authors selected patients with and withoud CVDs hospitalizations, performing a case-control study. So this point should be clarified. Authors are not really confident with epidemiological methodology.

About statistical analysis, Authors stated to perform PS matching in order to reduce possible biases. Authors should report baseline characteristics of total population before and after PS matching with relative standardized mean difference. It is absurd that in table 1, after matching procedure, several covariates remain statistically different between study groups. THe matching procedure totally failed.

About regression analysis, if Authors performed an appropriate PS matching, they don't need to perform a multivariate analysis. Logistic regression needs to be performed only with flu vaccination status adjusted for propensity score.

Introduction section was totally poor and need some improvements. No information about the type of vaccination performed was reported.

Reviewer 3 Report

Comments and Suggestions for Authors

The report by Zubair Akhtar et al concerns the impact of influenza vaccination on the prevention of cardiovascular disease in Australian patients over 50 years of age.

This is a case-control study with a 1:1 ratio. The study concludes in favor of cardiovascular protection by the influenza vaccine, which reinforces already known scientific evidence. The long latency between data collection (2017) and the publication of this report must also be justified. Another major limitation is the 1:1 ratio between cases and controls, which is very low. The disproportion between vaccinated controls (60%) and vaccinated cases (30%) also represents a significant imbalance.

Round 2

Reviewer 2 Report

Comments and Suggestions for Authors

Authors improved the manuscript

Reviewer 3 Report

Comments and Suggestions for Authors

I have not further requests